# *Paracoccidioides lutzii* Infects *Galleria mellonella* Employing Formamidase as a Virulence Factor

Elisa Dias Pereira[1☯], Thalison Rodrigues Moreira[1☯], Vanessa Rafaela Milhomem Cruz-Leite[1]*, Mariana Vieira Tomazett[1], Lana O'Hara Souza Silva[1], Daniel Graziani[2], Juliana Assis Martins[3], André Corrêa Amaral[3], Simone Schneider Weber[4], Juliana Alves Parente-Rocha[1], Célia Maria de Almeida Soares[1], Clayton Luiz Borges[1]*

1 Department of Biochemistry and Molecular Biology, Institute of Biological Sciences II, Federal University of Goiás, Goiânia, Brazil, 2 Multiuser Laboratory for the Evaluation of Molecules, Cells and Tissues, Federal University of Goiás, Goiânia, Brazil, 3 Laboratory of Nano&Biotechnology, Department of Biotechnology, Institute of Tropical Pathology and Public Health, Federal University of Goiás, Goiânia, Brazil, 4 Faculty of Pharmaceutical Sciences, Food and Nutrition, Federal University of Mato Grosso do Sul, Campo Grande, Brazil

☯ These authors contributed equally to this work.
* van-rafaela@hotmail.com (VRMC); clayton@ufg.br (CLB)

**Data Availability Statement:** All relevant data are within the manuscript and its Supporting Information files.

## Abstract

The formamidase (FMD) enzyme plays an important role in fungal thriving by releasing a secondary nitrogen source as a product of its activity. In *Paracoccidioides* species, previous studies have demonstrated the upregulation of this enzyme in a wide range of starvation and infective-like conditions. However, *Paracoccidioides lutzii* formamidase has not yet been defined as a virulence factor. Here, by employing *in vivo* infections using an *fmd*-silenced strain in *Galleria mellonella* larvae model, we demonstrate the influence of formamidase in *P. lutzii*'s immune stimulation and pathogenicity. The formamidase silencing resulted in improper arrangement of the nodules, poor melanogenesis and decreased fungal burden. Thus, we suggest that formamidase may be a piece composing the process of molecular recognition by *Galleria* immune cells. Furthermore, formamidase silencing doubled the observed survival rate of the larvae, demonstrating its importance in fungal virulence *in vivo*. Therefore, our findings indicate that formamidase contributes to *Galleria*'s immune incitement and establishes the role of this enzyme as a *P. lutzii* virulence factor.

## Author summary

The study of fungal neglected pathogens is of significant importance for the elucidation of the mechanisms underlying the diseases they cause. The *Paracoccidioides lutzii* fungus is a pathogen endemic to Latin America, which causes disease in vulnerable portions of society. To cause the disease, the fungal cells must overcome the immune system and obtain nutrients that are typically withdrawn by the host. In this regard, different molecules are crucial for the establishment of infection. The formamidase enzyme plays a role in the survival of the fungus *P. lutzii*. Our study, presented below, demonstrates that formamidase silencing affects the immune response of *G. mellonella* larvae to the fungus, resulting in

**Funding:** This work was majorly financed by Conselho Nacional de Desenvolvimento Científico e Tecnológico (CNPq) through CLB grant number 408042/2021-4. At the basal level, the techniques performed were also supported by Fundação de Amparo à Pesquisa do Estado de Goiás (FAPEG), Instituto Nacional de Ciência e Tecnologia (INCT-IPH-FAPEG),; grant number INCT-IPH-FAPEG 201810267000022 to CMAS. CLB has a fellowship from Conselho Nacional de Desenvolvimento Científico e Tecnológico (CNPq- 308237/2022-6). EDP, TRM and JAM receive salary from Coordenação de Aperfeiçoamento de Pessoal Nível Superior (CAPES). The funders had no role in study design, data collection and analysis, decision to publish, or preparation of the manuscript.

**Competing interests:** The authors have declared that no competing interests exist.

impairment of nodule formation, diminished melanin production, and a reduction in fungal burden. Furthermore, silencing formamidase expression in *P. lutzii* doubles the survival rate of larvae infected by the fungus, showing its importance for these yeast cells to infect this animals.

## Introduction

The genus *Paracoccidioides* comprises all the species of paracoccidioidomycosis (PCM) agents [1]. It is widely distributed throughout South America [2,3], and Brazil presents the highest prevalence in that continent [4]. Once spores or mycelial fragments have been inhaled, the host's body temperature triggers a morphological switch to the pathogenic yeast form [1].

During this initial contact, the human innate immune response ideally control the invasion through macrophages and neutrophils [5]. Phagocytosis is then a strategy usually aimed at killing microbes through acidic, oxidative [6,7] and nutritional stress [8]. However, as documented in *Histoplasma capsulatum* [9], *Cryptococcus neoformans* [10], *Candida albicans* [11], and *Paracoccidioides brasiliensis* [12], some fungal pathogens carry molecular mechanisms to overcome cellular response within the phagolysosome or granulomes.

During the yeast-host interaction, carbon and nitrogen sources are a crucial role in fungal thrive [13]. However, these sources are intentionally scarce in the host environment. They are necessary for energy metabolism and protein biosynthesis. Therefore, once a yeast is challenged by macrophage nutritional depletion, it must obtain nitrogen and carbon from non-preferred sources to survive and multiply [14]. In this context, the mechanism of nitrogen catabolite repression (NCR) plays an important role in regulating the scavenging, uptake and metabolism of non-preferred nitrogen sources in fungal pathogens [15]. The NCR mechanism was firstly explored in *Aspergillus nidulans* through investigations of ammonium, glutamine, and glutamate assimilation [15]. Studies conducted in *Saccharomyces cerevisiae* [16] and *Neurospora crassa* [17] have highlighted the role of this regulation in model fungi. In *A. nidulans*, the major regulator of NCR is the transcription factor AreA [18]. As a global acting modulator, AreA induces a set of various fungal genes related to metabolism of secondary nitrogen sources [19,20]. One of the NCR-modulated proteins is the formamidase (FMD) (E.C. 3.5.1.49). The FMD enzyme is a virulence factor of *Aspergillus* that exhibits *in vitro* hydrolase activity on formamide, releasing formate and ammonium [21]. These products may subsequently serve as wounding factors and alternative nitrogen sources [22].

Previous studies demonstrated that recombinant formamidase significantly reacts with sera from patients with PCM, but not with uninfected ones [23]. Additionally, this enzyme was found to be the most expressed protein in the yeast proteome during infection models, as well as in protein extract of isolated conidia [24–27]. Complementarily to these findings, the detection of formamidase as an exoantigen [28] and its presence in both the cytoplasm and cell wall [29] suggest that it plays a crucial role in host-pathogens interactions.

The *Galleria mellonella* (Lepidoptera: Pyralidae) *in vivo* model is an important tool for investigating pathogenic microorganisms, including bacteria and fungi [30–31]. Their immune system shares some mechanisms with that of mammals making them a viable model for this purpose [7,32]. One such response is that of differentiated hemocytes which resemble some mammalian defense cells. The four types of this cells function collectively to fight microbial proliferation [33]. For example, oenocytes and spherules transport and release molecular components, such as pro-phenoloxydase (PPO), the first protein in melanogenesis cascade [34]. In addition to these, plasmatocytes and granulocytes, which are adherent cells, are associated with phagocytosis, capsule formation, and nodulation [35]. Within the cellular response,

nodules in *Galleria*'s resemble granulomes, which are structures created by cell aggregation that ideally isolate and kill fungi through immune enzymes and clotting factors [36]. A nodule is created when the number of fungal cells exceeds the phagocytosis capacity of a single cell. The conglomerates increase in size as new hemocytes attach to the surface.

During the later stages of nodulation, melanogenesis begins due to the degranulation of vesicles containing PPO enzyme by oenocytes [34]. The immune activity of melanin has been widely investigated in insects genus such as *Drosophila* [37], *Aedes* [38], and *Galleria* [39]. This pigment mediates fungal killing mainly through oxygen and nitrogen reactive species and cytotoxic intermediates [34]. The feasibility of *Galleria*'s *in vivo* model was demonstrated for *P. lutzii* [40], and *P. brasiliensis* within the *Paracoccidioides* genus [41]. Additionally, a comparison of both species' virulence was conducted in this model [42].

Thus, we hypothesize that formamidase serves as both a molecular trigger for *Galleria*'s immune response and a virulence factor of *P. lutzii* contributing to yeast pathogenesis. In this study, we aim to demonstrate the influence of this enzyme on nodulation and mortality of *P. lutzii* after *in vivo* infection, building upon previous research. To achieve this, we injected an *AS-fmd* that had been silenced using antisense technology [43] into *G. mellonella* larvae. We then used histotechnological methods to stain and analyze the larvae's tissues and plotted a Kaplan-Meyer survival curve (Fig 1).

## Results and discussion

### Formamidase contributes to nodular melanogenesis

Melanogenesis is a crucial mechanism for antimicrobial response in arthropods [37]. To investigate melanization patterns within the nodules, we stained tissue sections of larvae infected

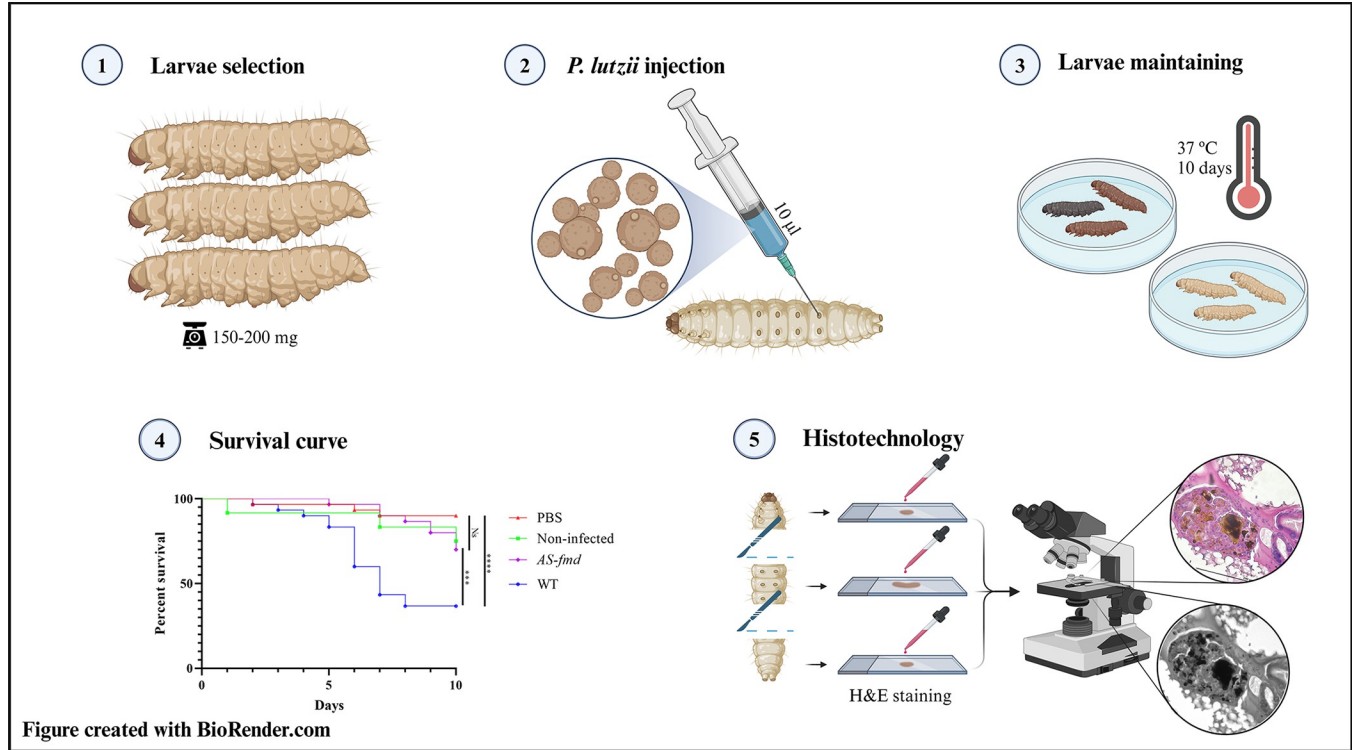

**Fig 1. Experimental workflow for larval infection and histotechnology.** – The larvae were infected with two strains of *P. lutzii* and the death events were assessed for ten days to build the Kaplan-Meyer survival curve. Histotechnological analyses were performed to investigate the effects causing larval death. Figure created with BioRender.com.

with both wild-type (WT) and silenced *P. lutzii* yeast cells (AS-*fmd*). To demonstrate the frequency and intensity of melanin spots, we split the blue channel of each image digitally [44]. No signs of melanization or nodulation were observed in the images obtained from both the PBS-injected and non-infected groups of larvae. Also, the peripheral plasmatocytes did not show any evidence of phagocytized content (S1 Fig). A significant difference in melanization pattern was observed between the wild-type and transformant infected groups of larva. Melanin indicators were present in all analyzed nodules from the wild-type-infected group, with higher intensity in certain sites. The pixel intensity of the images was evaluated digitally and demonstrated a notable discrepancy between the two groups of larvae. The group infected with the wild type exhibited a higher mean pixel count, indicating an elevated level of melanization (S2 Fig and S1 Table). The pixel intensity of the images was evaluated digitally and demonstrated a notable discrepancy between the two groups of larvae. The group infected with the wild type exhibited a higher mean pixel count, indicating an elevated level of melanization (S2 Fig and S1 Table). These sites involved the yeast cells beyond their cell walls, suggesting a major composition of immune melanin (Fig 2A–2C). The blue-channel images reveal multiples dark sites that suggest higher levels of melanin (Fig 2G–2I). These melanized sites make it difficult to accurately identify the exact shape of the yeasts. Conversely, the tissues collected from AS-*fmd* infected larvae display clear yeast shapes with light-brown regions that are typically confined to the cell wall (Fig 2D–2F). The reduced intensity of dark-brown sites suggests a cellular response with lower melanogenesis compared to the wild-type (Fig 2J–2L). These patterns of accumulation could also indicate presence of fungal melanin [45], that requires additional investigation.

Melanogenesis depends on the recruitment of oenocytes, which is driven by activated plasmatocytes [46]. Subsequently, oenocytes initiate the melanin synthesis cascade by delivering granules of PPO within the nodules. Melanin has been extensively proven to have antimicrobial properties through inference studies with PPO mutants [47,48]. Recently, its fungicidal activity was investigated in detail further supporting its role in microbial killing [39]. Furthermore, studies on *Aspergillus terreus* and *Klebsiella* strains demonstrated a correlation between an increase in melanin content and higher mortality rates of infected larvae [49,50]. Therefore, the observed impairment of melanogenesis in AS-*fmd* infected nodules indicates lower virulence of this strain as well as incomplete immune activation.

### The arrangement of the nodules is influenced by formamidase

In insects, the nodular structure is usually well defined. It begins with an ordered recruitment of hemocytes based on pattern recognition. These granuloma-like bodies are mainly composed of phagocytic granulocytes and secretory cells [51]. To test the level of formamidase impact in the cell disposition of the nodules, we examined the overall nodule structure of the two groups of larvae. The wild-type infected group showed a finely organized structure (Fig 3A). Inside the nodule, the presence of brown-dark melanization sites, yeast cells and hemocytes–most likely granulocytes–is evident (Fig 3B). A well-defined basement membrane-like (BM-like) plasmatocytes clearly separates the nodule from the adjacent tissue (Fig 3C). Nodular isolation may represent an appropriate immune response to the pathogen. This sequestration of yeast cells toward the center keeps them closer to the aggregate of hemocytes and protects the surrounding tissue. In contrast, the mutant infected group showed a sparse distribution of yeast cells throughout the nodular structure (Fig 3F). In addition, the BM-like layer appears to be considerably thick with almost no morphological distinction from the granular inner cells (Fig 3H). Taken together, these results indicate that the nodule structure and hemocyte disposition are highly affected by formamidase levels.

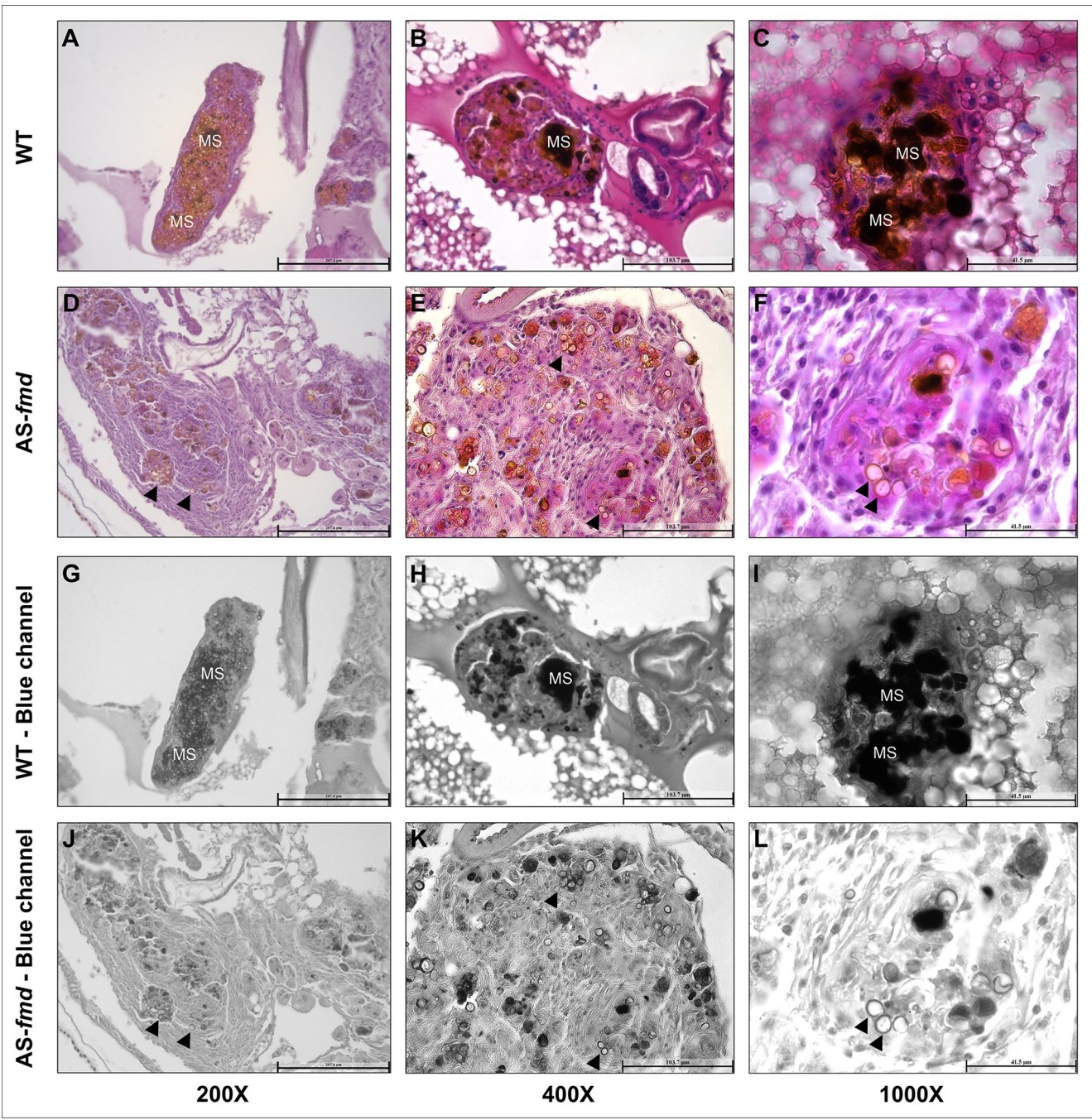

**Fig 2. –Melanization patterns of larvae infected with wild-type and AS-*fmd* P. lutzii strains.** Tissue slides displaying nodules from three different specimens of *G. mellonella* infected with wild-type (A-C) and AS-*fmd* strains (D-F) then stained with HE. The images of the blue channel split highlight the melanin spots (G-L). **A-C)** The three different nodules enclosing *P. lutzii* wild-type cells show well-defined structures, tight cellular organization, and high intensity of dark regions derived from melanin stain (MS), demonstrating increased levels of melanin. **D-F)** The mutant cells of *P. lutzii* are encompassed by three distinct nodules that have softly defined structures, loose cellular disposition, and clear-brown regions, indicating low levels of melanin. **G-L)** The wild-type and AS-*fmd* images after digital splitting of the blue channel to evidence melanin. The darker regions of each image indicate high melanized sites, while the clear-brown regions bordering the cell wall indicate low melanized sites (MS). AS-*fmd* images show yeast cells with a slightly darker color delineating the cell wall (arrowheads).

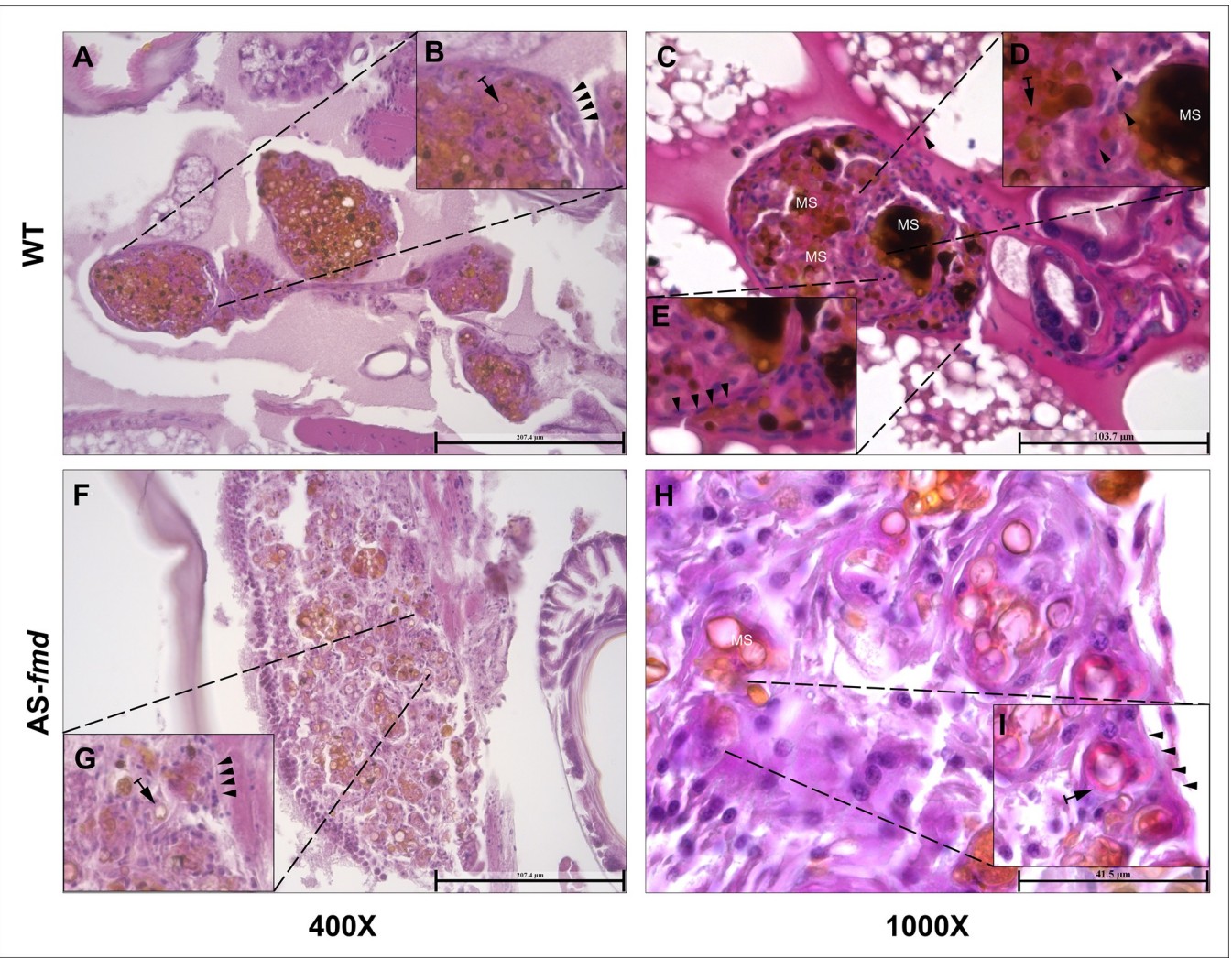

**Fig 3. −Structure of the nodular tissues from infected larvae.** Tissue slides from four different animals stained with HE. Visualization with 400x and 1000x magnification. **A and C)** The structures of nodules from two different specimens infected with the wildtype strain are finely arranged with a tight cell disposition. **B, D and E)** There is a precise BM-like layer of plasmatocytes strictly delimiting the nodular boundaries (arrowheads), and numerous yeasts surrounded by defense cells (arrows). **F and H)** Tissue from two different specimens infected with the AS-*fmd* strain. Both nodules exhibit a loose arrangement and cellular sparseness. **G and I)** There is a poorly defined and incontiguous BM-like layer (arrowheads) with impaired plasmatocyte discrimination. Yeast cells are present in low numbers surrounded by hemocytes (arrows).

In insects, nodule formation begins with the recognition of pathogen associated molecular patterns (PAMPs) and damage associated molecular patterns (DAMPs) by pattern recognition receptors (PRRs). The three main PRRs involved in this process are the Toll Receptors (TR), Janus Kinase/Signal Transducers and Activators of Transcription (JAK/STAT) and Immune Deficiency (Imd) pathways [52]. The Imd pathway is usually associated with DAMPs and Gram-negative bacteria signaling, while TR and JAK/STAT recognize fungal, viral, and Gram-positive bacteria molecules [53–55]. Once triggered, the PRRs signal the activation of the immune response. Starting with the attachment of granulocytes to the yeast cells, the nodule reaches the classical structure through granulocytes degranulation and agglutination. Subsequently, it leads to nodular growth due to plasmatocyte adhesion and spreading [56].

Previous studies have described enzymatic activity playing a role of DAMP in *Drosophila* species [57]. Moreover, PRRs are typically capable of recognizing a wide variety of microbial

PAMPs, including peptides, glycans and saccharides [36,55]. Here, we propose that the decrease in FMD at protein level notated in AS-*fmd* [43] along with the nodular changes discussed is a step towards incomplete nodulation [58]. These findings suggest that formamidase may be involved in the process of triggering the immune response. Further investigation is required to elucidate the specific role of FMD in the generation of the immune response of *G. mellonella* though.

## Formamidase is associated with increased fungal burden in the nodules

Tissue damage and scavenging of secondary nutrient sources are essential aspects for fungal survival during nodular isolation. To investigate whether formamidase silencing affects fungal load inside the nodules, we compared tissue slides stained with HE from both groups of larvae. The wild-type infected group showed an elevated fungal burden within the nodules (Fig 4). Despite the proper constitution of the BM-like layer, there is a predominance of yeast cells instead of hemocytes in the middle of the nodules (Fig 4A and 4B). In contrast, the AS-*fmd* showed a lower fungal burden and a higher recurrence of hemocytes within its nodules compared to the wild-type (Fig 4C and 4D).

The ability of *Paracoccidioides* to survive and even multiply inside phagocytes is well known [12,59]. A wide range of adaptative mechanisms are employed by the fungus to succeed in this harsh environment [6]. Previous studies demonstrated the importance of formamidase for *P. lutzii* survival in macrophages [43] and its potential as a virulence factor. In addition, its significant expression in *in vitro*, *ex vivo* and *in vivo* infections suggested formamidase as a crucial component of nitrogen metabolism inside the host [60]. Consistently, our findings indicate that formamidase operates by improving yeast endurance and/or multiplication inside the nodules.

## Formamidase is a virulence factor of *P. lutzii*

Our group previously demonstrated that formamidase contributes to an increased survival rate of *P. lutzii* phagocyted by murine macrophages [43]. These results indicated the role of formamidase in the meaningful fungal resistance to innate immune response. To assess if formamidase contributes to fungal pathogenicity, we employed an *in vivo* infection of AS-*fmd* in *G. mellonella* following previous methods [40]. As controls we set a group of larvae injected with phosphate-buffered saline (PBS) (Fig 5, red triangles) and another non-injected at all (Fig 5 green squares). An elevated survival rate was observed in the PBS group (90%). It demonstrates a minor impact of the trauma caused by needle insertion and the liquid inoculation itself in larval death. Despite the decrease compared to the saline group, non-infected larvae showed a similarly high rate of survival (75%), confirming little effect of the management conditions on the animals. This reduction in survival rate is explained by two death events caused by starvation inherent to the experiment. Between the infected groups, we found a survival rate of the wild-type group significantly lower than both controls' (Fig 5). Inoculation of the wild-type strain culminated in only 35% survival (Fig 5, blue circle), highlighting the substantial pathogenicity of *P. lutzii* to *Galleria*. Conversely, as we expected, the survival rate of the AS-*fmd* infected animals was significantly higher than that of wild-type (70%) (Fig 5, purple diamond). Additionally, the mutant strain showed no significant difference in survival rate from those seen in control larvae (Fig 5).

Formamidase may support nutrient intake in scarce environments, raising fungal fitness. *P. brasiliensis* expressed formamidase in a ten-fold greater rate during broad nutritional starvation [61]. Interestingly, formamidase is induced in *Paracoccidioides* during iron deprivation [62], carbon starvation [63], and repressed in *Histoplasma capsulatum* under copper

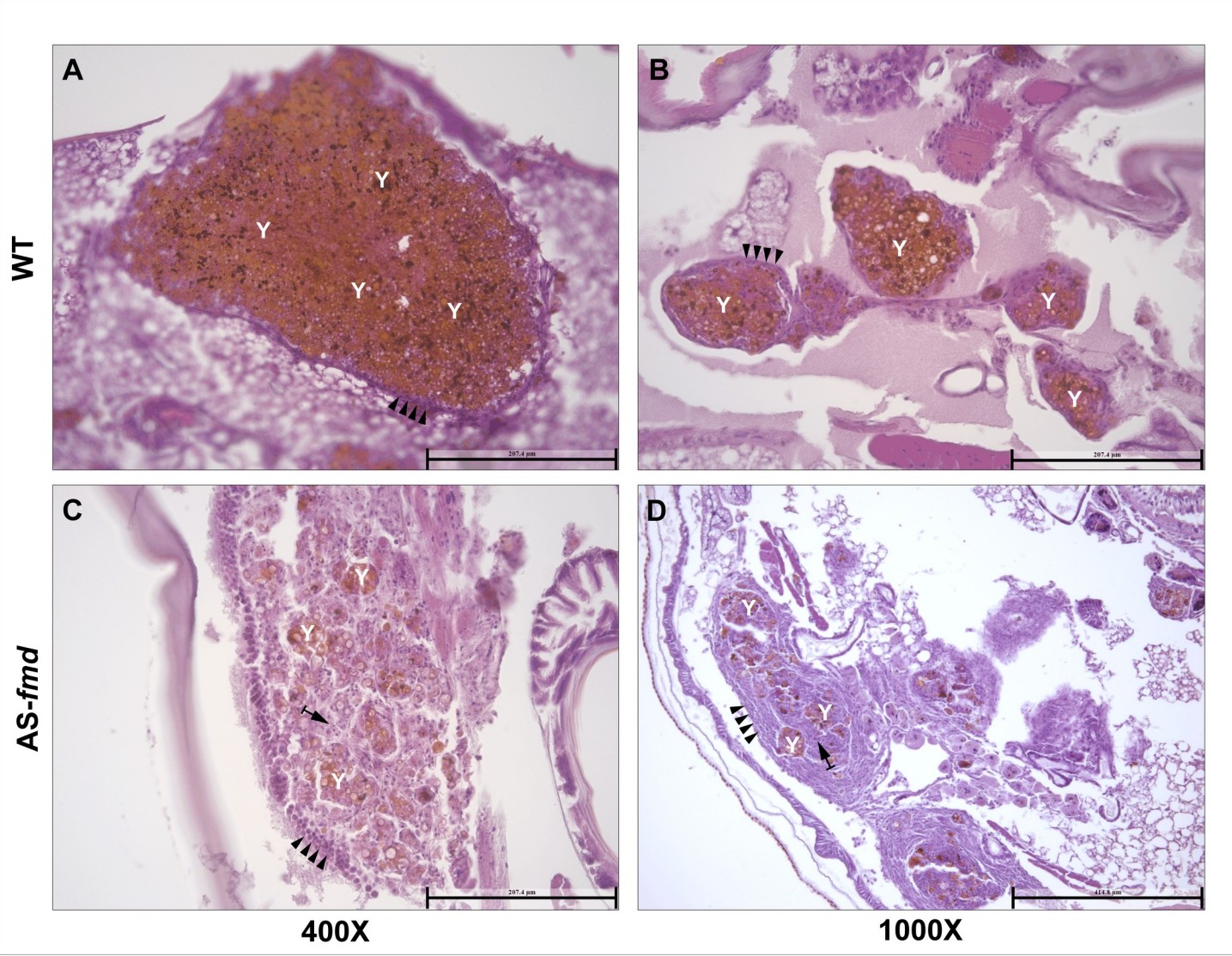

**Fig 4. −Fungal burden inside the nodules from both groups.** Fungal burden was assessed inside the nodules of both groups. Tissue slides from the wild-type and AS-*fmd* group were HE-stained and viewed at 400X magnification. **A and B)** Tissue slides of nodules from distinct animals displaying yeast predominance (Y) in their interior. The BM-like monolayer of plasmatocytes is continuously surrounding the nodule and accurately defined (arrowheads). **C and D)** Nodules from two distinct animals showing a predominance of hemocytes (arrow). The BM-like layer is composed of several hemocytes and is disconnected in some areas (arrowheads), indicating improper formation. The yeast cell content is found in small clusters, demonstrating a low fungal load (Y).

overabundance [64]. However, the role FMD in these conditions is still unclear. Studies conducted on wild-type strain of *P. brasiliensi*s showed a six-fold increase in formamidase expression when grown on minimal medium compared to complete medium [61]. Additionally, *P. lutzii* upregulates formamidase during nitrogen depletion, suggesting its function in the acquiring this macronutrient [22]. These findings are consistent with previous studies of *A. nidulans* [21,65]. Despite there is no conclusion about the *in vivo* substrate of this enzyme [66] it has substrate looseness that could be operating on unspecific targets such as butanamide, acetamide, propenamide, and N-formyl-kynurenine. Thus, FMD is a crucial component to the response of fungal pathogens to nitrogen depletion due to ammonium production. These findings demonstrate that formamidase is especially required during nitrogen exhaustion to acquire and metabolize alternative sources of this nutrient.

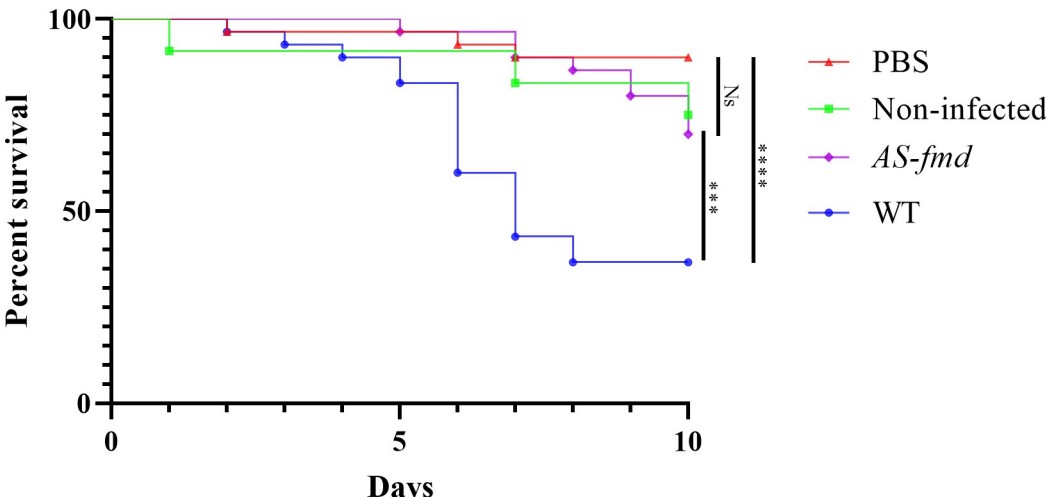

**Fig 5. –Survival curve of *G. mellonella* infected with *P. lutzii* strains.** Four groups of larvae were injected with PBS (red triangle), non-infected (green square), AS-*fmd* (purple diamond), and wild-type (blue circle) suspensions. Death events were recorded and counted over a 10-days period. The collected data was plotted, and a Kaplan-Meyer curve was generated to analyze the results. The statistical difference between pairs of groups was assessed using the Gehan-Breslow-Wilcoxon test. The survival rates of the PBS and non-infected groups were not significantly different (p-value = 0.2303), serving as controls. The wild-type and PBS groups were highly different (p-value < 0.0001). The AS-*fmd* showed no significant difference when compared to either PBS (p-value = 0.095) or non-infected larvae (p-value = 0.8607). However, the AS-*fmd* and wild-type groups were significantly different (p-value 0.0006). The figure was created using GraphPad Prism 8 software. We use (****) to indicate p-value < 0.0001; (***) for p-value < 0.001; and (Ns) for non-significant difference.

Several studies demonstrated the significance of amidases in the pathogenicity of microbes. The role of ammonium produced by urea-specific amidase (urease, E.C. 3.5.1.5) in virulence has been well studied in bacterial species including *Helicobacter pylori*, *Proteus mirabilis*, and *Klebsiella pneumoniae* [67–69]. In *C. neoformans*, this outcome ammonium also serves as a nitrogen source and virulence factor along with growth signaling and yeast-yeast communication [70–72]. The same contributions of urease are observed in *Coccidioides posadasii* [73]. In addition, *C. albicans* has the ability of manipulate the phagosome acidic condition through ammonium extrusion [74]. In *Aspergillus fumigatus*, inhibition of the BET epigenetic virulence regulator resulted in reduced protein expression of a putative formamidase [75]. The BET protein family typically modulates multiple important virulence factors in fungi [76]. Within the same species, resistance to azoles is associated with increased formamidase abundance [77]. FMD was also identified as a component of extracellular vesicles of *H. capsulatum* [78]. These vesicles are often associated with fungal pathogenesis and host inflammation [79]. Therefore, in addition to nutrient scavenging, FMD may also exert its role as a virulence factor through tissue damage, fungal growth modulation, and cell signaling (Fig 6).

In *Paracoccidioides* species, FMD was identified in the cell wall, cytoplasm, and secretome of mycelia and yeast [29,80,81], indicating its expression and necessity for the fungi in different conditions. Additionally, its significant reactivity to PCM patients' sera [23] and exoantigen status [28], highlight the potential of FMD to be efficiently recognized by the immune response. Moreover, its transcripts are highly induced in return to yeast exposition to human blood [82], *in vitro* and *in vivo* infections [60] and recovered from mice lungs [25], suggesting it as a step in host-colonization. Therefore, our findings are consistent with previous knowledge and demonstrate that FMD is a virulence factor that influences survival and innate immune response of *Galleria* larvae.

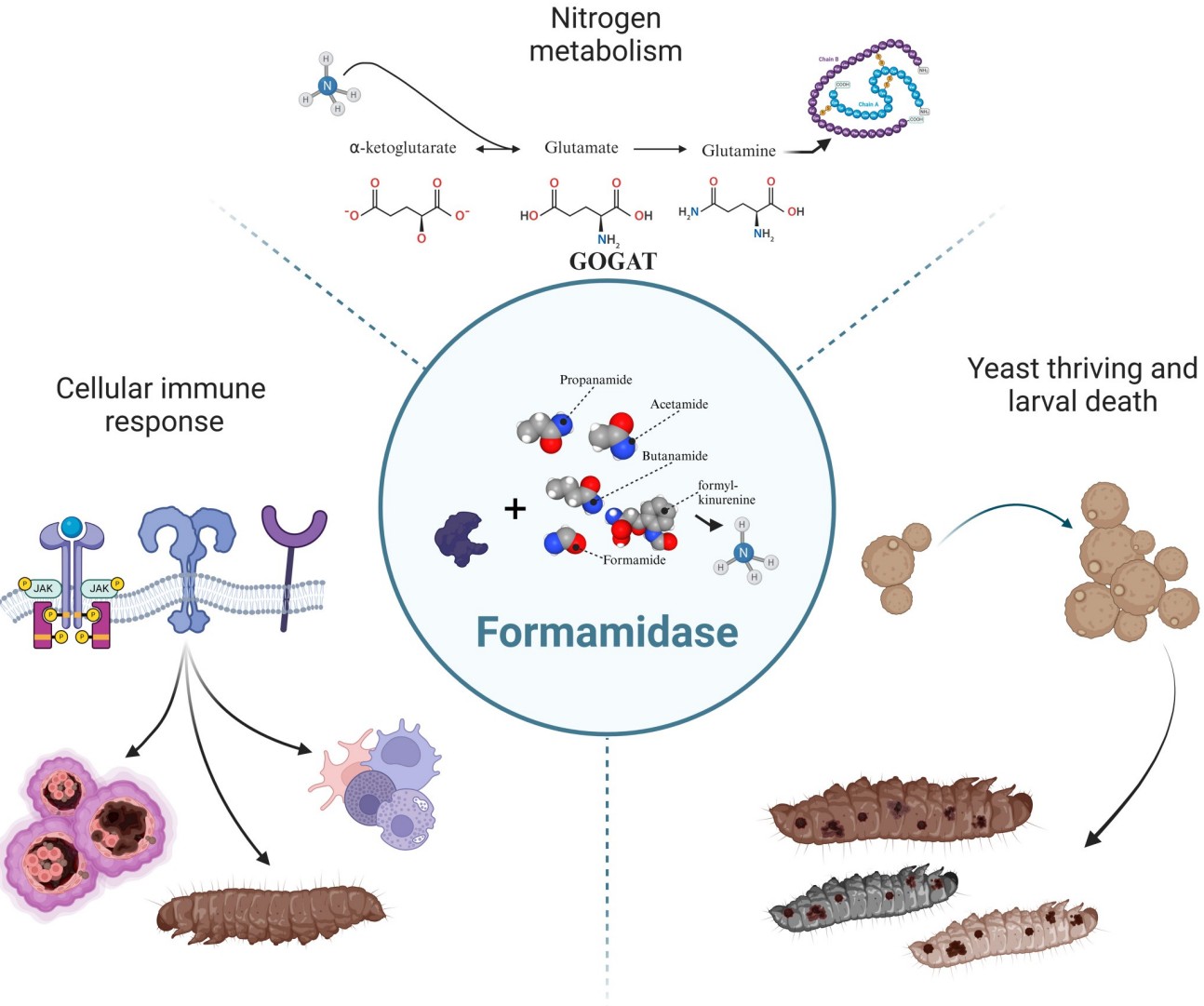

**Fig 6. –Probable impacts of formamidase during *P. lutzii* infection in *G. mellonella* model. Top)** The FMD enzyme is closely related to ammonium insertion into the glutamate and glutamine synthesis. It serves then as a source of these amino acids to the yeast protein metabolism, increasing fungal survival or even multiplication. **Right)** The generated ammonium may act as a cellular signal stimulating yeast-yeast communication to growth. This role played by the FMD-generated ammonium is a step towards larval death. **Left)** Immune recognition by PRRs induces nodulation, humoral response, fungal burden, and body melanization. Figure created with BioRender.com.

## Material and methods

### *Paracoccidioides lutzii* and *Galleria mellonella* maintenance

The larvae were maintained under diet of honeybee wax and pollen at 25°C until reaching 150–200 mg to start infection. Cream colored larvae were selected and placed in petri dishes for 24 hours without feeding at 25°C. *P. lutzii* wild-type (wild-type) cells were cultured in non-selective media Brain Heart Infusion (BHI) supplemented with 1.1% (w/v) glucose. The silenced strain (*AS-fmd*) was kept in selective solid BHI containing hygromycin B (75 μg/mL) supplemented with 1.1% (w/v) glucose. Both strains were cultivated at 36°C for 72 hours. Prior to the infection, both *P. lutzii* strains were inoculated in liquid BHI for 48 hours at 150 rpm. Finally, yeast cells were recovered for the *in vivo* infection.

## Construction of the AS-*fmd* silenced *P. lutzii* strain

The *P. lutzii* formamidase gene (*fmd*) was silenced by the antisense RNA (aRNA) technique in conjunction with *Agrobacterium*-mediated transformation (ATMT), as previously described [43]. Briefly, forward (5′ CCGCTCGAGCGGCTTGCATAACCGCTGGCATC 3′) and reverse (5′ GGCGCGCCTCGTCGGCGGAATCGTTATT 3′) oligonucleotides were designed to generate an antisense fragment (AS) of the *P. lutzii* formamidase gene. A binary plasmid (based on pUR5750) containing the AS-*fmd* and a gene for resistance to hygromycin B was then randomly inserted into the *P. lutzii* genome. Integration of this transfected cassette into the fungal genome was confirmed by conventional PCR using pUR5750 oligonucleotides.

## *In vivo* infection *of P. lutzii strains* in *G. mellonella*

Both wild-type and silenced strains were harvested by centrifugation, washed three times, and then resuspended in phosphate-buffered saline (PBS). The cells underwent repeated mechanical force using a 5 mL syringe and filtration through a 40 μm nylon filter to diminish cellular clusters. The ventral region of the insect was sterilized with 70% ethanol prior to the injection of cells. Using a Hamilton syringe (Sigma-Aldrich, St. Louis, MO, USA), we injected a yeast suspension containing $5\times10^6$ cells in a volume of 10 μl into the last pro-leg on the left side of the larvae [40]. During the ten-day period, the larvae were maintained in Petri dishes at a temperature of 37˚C. The dead larvae were identified by steadiness after stimulation with forceps and then removed from the plate. The larvae in the inoculation control group were injected with 10 μl of PBS. For the environmental control group, the larvae did not undergo injection. Infections were carried out with a total of 30 larvae per group. [42]. All the death events were plotted to build a Kaplan-Meyer curve and underwent the Graham-Breslow-Wilcoxon and the Mantel-Cox statistic test with 95% confidence interval. The data analysis was performed on GraphPad Prism8 (www.graphpad.com).

## Production and staining of the tissue slides

The two darker specimens from each group were measured and sectioned into three pieces of similar length. They were then placed in the tissue processor Leica TP1020 (Leica Biosystems, Germany), for 12 hours to undergo gradual ethanol and xylol baths before being embedded in paraffin. The sections were recovered from paraffinization and carefully arranged into solid paraffin blocks. The blocks were trimmed using the Leica RM2255 microtome (Leica Biosystems, Germany) with 16μm-width cuts until a full-faced section with all three tissue parts was visible. Following a 10-minute ice-cold water bath, the blocks were slowly sectioned at 8μm with uniform rotations. The resulting sections were then placed in a warm water bath using a small brush and transferred onto glass slides. The tissue sections were subjected to xylol-mediated deparaffinization, ethanol hydration, and staining with hematoxylin and eosin. The slides were dehydrated, fixed, and covered with a glass cover and Entellan™ mounting medium (Merck, Germany). All the images were then assessed using light microscopy with the Leica DM6 B microscope and the Leica DFC7000T camera, and processed with the software LAS X (Leica, Germany). The three image channels were split with the ImageJ software (NIH, United States), and the blue color channel selected to evidence melanized spots [44]. The melanin intensity of the spots was evaluated through digital analysis of the pixel intensity of six images from each infected group of larvae. Subsequently, a two-tailed unpaired Student's t-test was conducted on the data with 95% confidence.

## Supporting information

**S1 Fig. Tissue slides from control groups HE and GG stained. A-C)** Tissue slides from non-infected group stained with HE, highlighting *Galleria*'s cuticle (arrows), peripheral

plasmatocytes (arrowhead), adipose cells (dashed arrows), and hemolymph (asterisk). There is no evidence of melanization, nodulation or yeast cells. **D-E)** Tissue sections stained with silver Gomori-Grocott method confirming absence of yeast structures. The *Galleria's* cuticle (arrows), adipose tissues (dashed arrows), and peripheral plasmatocytes (arrowhead) are shown in evidence. **G-I)** Tissues from the PBS-injected group stained with HE highlights larvae's hemolymph (asterisk). **J-L)** Gomori-Grocott stained tissues from the PBS-injected group of larvae. It emphasizes the absence of fungal structures. *Galleria*'s structures such as cuticle (arrows), adipose tissue (dashed arrow), and hemolymph (asterisk) are apparent.
(TIF)

**S2 Fig. Pixel intensity of the melanin spots from each infected group of larvae.** The graphical representation of the pixel measurements obtained from nodular melanin spots of the wild type and AS-*fmd* strains is presented herewith. The statistical significance of these data was evaluated using a Student's t-test with the aid of GraphPad Prism8 software. The two asterisks (\*\*) denote a p-value $< 0.001$.
(TIF)

**S1 Table. Table presenting the pixel measurements of melanin spots located within the nodules of wild type and AS-fmd-infected larvae.** A total of six images of each group underwent the process using the FIJI software. Three of the measured images are those presented in Fig 2.
(XLSX)

**S2 Table. Data of death events used in survival curve.**
(XLSX)

## Author Contributions

**Conceptualization:** Thalison Rodrigues Moreira, Juliana Alves Parente-Rocha, Clayton Luiz Borges.

**Data curation:** Thalison Rodrigues Moreira.

**Formal analysis:** Thalison Rodrigues Moreira.

**Funding acquisition:** Célia Maria de Almeida Soares, Clayton Luiz Borges.

**Methodology:** Elisa Dias Pereira, Thalison Rodrigues Moreira, Vanessa Rafaela Milhomem Cruz-Leite, Mariana Vieira Tomazett, Lana O'Hara Souza Silva, Daniel Graziani, Juliana Assis Martins.

**Project administration:** Clayton Luiz Borges.

**Resources:** André Corrêa Amaral, Célia Maria de Almeida Soares.

**Supervision:** André Corrêa Amaral, Simone Schneider Weber, Juliana Alves Parente-Rocha, Célia Maria de Almeida Soares, Clayton Luiz Borges.

**Validation:** Thalison Rodrigues Moreira.

**Visualization:** Thalison Rodrigues Moreira.

**Writing – original draft:** Thalison Rodrigues Moreira, Juliana Alves Parente-Rocha, Clayton Luiz Borges.

**Writing – review & editing:** Elisa Dias Pereira, Thalison Rodrigues Moreira, Lana O'Hara Souza Silva, Simone Schneider Weber, Juliana Alves Parente-Rocha, Clayton Luiz Borges.

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
