## [Decision Letter · Decision Letter 0]

31 May 2024

Dear Professor Borges,

Thank you very much for submitting your manuscript "Paracoccidioides lutzii Infects Galleria mellonella Employing Formamidase as a Virulence Factor" for consideration at PLOS Neglected Tropical Diseases. As with all papers reviewed by the journal, your manuscript was reviewed by members of the editorial board and by several independent reviewers. In light of the reviews (below this email), we would like to invite the resubmission of a significantly-revised version that takes into account the reviewers' comments. 

We cannot make any decision about publication until we have seen the revised manuscript and your response to the reviewers' comments. Your revised manuscript is also likely to be sent to reviewers for further evaluation.

Sincerely,

Joshua Nosanchuk, MD

Section Editor

Joshua Nosanchuk

Section Editor

Reviewer's Responses to Questions

**Key Review Criteria Required for Acceptance?**

**Methods**

-Are the objectives of the study clearly articulated with a clear testable hypothesis stated?

-Is the study design appropriate to address the stated objectives?

-Is the population clearly described and appropriate for the hypothesis being tested?

-Is the sample size sufficient to ensure adequate power to address the hypothesis being tested?

-Were correct statistical analysis used to support conclusions?

-Are there concerns about ethical or regulatory requirements being met?

Reviewer #1: The objectives of the study are clearly articulated, focusing on the role of P. lutzii formamidase during interaction with G. mellonella. The study design appears appropriate for addressing the objectives. The use of G. mellonella larvae as a model organism to study fungal-host interactions is well-founded. The manuscript provided information regarding the sample size. The authors need to clarify whether appropriate statistical analyses were conducted to support the conclusions, particularly in relation to the results presented in Figure 3. It is essential to specify the statistical methods used, and whether any comparisons between groups were made to validate the findings.

Reviewer #2: Are the objectives of the study clearly articulated with a clear testable hypothesis stated?

-Is the study design appropriate to address the stated objectives? YES

-Is the population clearly described and appropriate for the hypothesis being tested? YES

-Is the sample size sufficient to ensure adequate power to address the hypothesis being tested? YES

-Were correct statistical analysis used to support conclusions? YES

-Are there concerns about ethical or regulatory requirements being met? YES

Additional comments on MATERIALS AND METHODS

Did the authors perform a growth curve of the control and the silenced (AS-find) strain for growth rate control purposes?

**Results**

-Does the analysis presented match the analysis plan?

-Are the results clearly and completely presented?

-Are the figures (Tables, Images) of sufficient quality for clarity?

Reviewer #1: The results are well-presented, but there are points that need more detailed explanations. Additional information is needed regarding the construction of the AS-fmd strain and further exploration of the findings in Figures 1 and 2.

Reviewer #2: -Does the analysis presented match the analysis plan? YES

-Are the results clearly and completely presented? YES

-Are the figures (Tables, Images) of sufficient quality for clarity? YES

Additional comments on RESULTS AND DISCUSSION

Fig. 1. Could the authors measure the intensity of the spots and transform it to numeric data?

Also, is the formation of these melanin spots correlated to the viability of the fungus? What happens if the authors inject a non-viable wild type to G. mellonella?

Could the authors perform any immunohistochemistry to accurately determine the arrangement of fungal cells in G. mellonella tissues upon infection?

It is not possible to accurately visualize the number of fungal cells comparing both strains (WT x AS-find silenced). Have the authors perform any CFU experiments to determine the viable fungal loads upon infection, as melanogenesis itself has been proven to have antimicrobial activities as the authors themselves stated.

What is the average of nodular tissues observed in WT x AS-find silenced infected G. melonnella?

Could the authors include the data of phagocytosis and survival rates of WT x AS silenced strains upon interactions with haemocytes?

What is the melanization pattern of the observed global larvae upon infection comparing both groups? Could the authors also perform a curve of melanization over time, for both groups and the controls?

**Conclusions**

-Are the conclusions supported by the data presented?

-Are the limitations of analysis clearly described?

-Do the authors discuss how these data can be helpful to advance our understanding of the topic under study?

-Is public health relevance addressed?

Reviewer #1: The conclusions are supported by the presented data, but it would be interesting to explore the hypotheses highlighted in Figure 5 further to ensure they are fully substantiated by the results. The authors should address the public health relevance in detail.

Reviewer #2: -Are the conclusions supported by the data presented? YES

-Are the limitations of analysis clearly described? YES

-Do the authors discuss how these data can be helpful to advance our understanding of the topic under study? YES

-Is public health relevance addressed? YES

**Editorial and Data Presentation Modifications?**

Reviewer #1: (No Response)

Reviewer #2: (No Response)

**Summary and General Comments**

Reviewer #1: Pereira et al. explore the role of P. lutzii formamidase during interaction with G. mellonella larvae. The experiments were well-executed, but some concerns need to be addressed:

1. Revise the reference formatting to correct inconsistencies throughout the manuscript.

2. Include additional information regarding the construction of the AS-fmd strain in the methods and/or results sections.

3. The authors discuss the following regarding the images in Figure 1: “The reduced intensity of dark-brown sites suggests a cellular response with lower melanogenesis compared to the wild-type (Fig. 1, J-L). These patterns of accumulation may also indicate the presence of fungal melanin (44), which only became visually apparent due to the severe nodular melanin reduction. Further analysis is needed to confirm this idea.” What specific analyses could be conducted to confirm this hypothesis? It is important to explore this possibility further in the text.

4. The results presented in Figure 2 are insufficient to confirm the role of FMD as a DAMP during fungal-host interaction. This hypothesis needs to be explored in more detail. Are there other evidences that could support this specific role?

5. The authors conclude that formamidase is associated with increased fungal burden in the nodules based on the results in Figure 3. Did the authors perform any statistical analysis to compare the groups? What strategies should be used to quantify the observed differences?

6. The abstract highlights that the authors “suggest that formamidase is a crucial step in molecular recognition by Galleria immune cells.” Please revise this sentence considering the points discussed above.

Reviewer #2: Dear Editor,

The manuscript by Pereira et. al. explores the participation of the P. lutzii formamidase in a model of G. mellonella infection by performing a series of histological evaluations and survival experiments comparing a wild type strain x silenced strain (AS-find) of P. lutzii. This manuscript is a nice contribution to the field, but some comments need to be addressed before its consideration for publication.

Here is a point-by-point comments for every section

ABSTRACT

Page 5, line 8. The authors should re-write this sentence. What do they mean, the participation of formamidase or its products?

INTRODUCTION

Some references are in numbered format and others (AUTHORS, year). The authors should always carefully review the whole document before submission. Also, the spaces between sentences and references.

What is the role of the ammonium generated by formamidase on the buffering of the pH? Please address.

Could the authors test the formamidase activity of these silenced strain (AS-find), in terms of cytoplasmic extracts and also exoantigens?

RESULTS AND DISCUSSION

Growth rate of both strains? Hygromycin plays any pressure on the AS-find strain?

RESULTS AND DISCUSSION

Fig. 1. Could the authors measure the intensity of the spots and transform it to numeric data?

Also, is the formation of these melanin spots correlated to the viability of the fungus? What happens if the authors inject a non-viable wild type to G. mellonella?

Could the authors perform any immunohistochemistry to accurately determine the arrangement of fungal cells in G. mellonella tissues upon infection?

It is not possible to accurately visualize the number of fungal cells comparing both strains (WT x AS-find silenced). Have the authors perform any CFU experiments to determine the viable fungal loads upon infection, as melanogenesis itself has been proven to have antimicrobial activities as the authors themselves stated.

What is the average of nodular tissues observed in WT x AS-find silenced infected G. melonnella?

Could the authors include the data of phagocytosis and survival rates of WT x AS silenced strains upon interactions with haemocytes?

What is the melanization pattern of the observed global larvae upon infection comparing both groups? Could the authors also perform a curve of melanization over time, for both groups and the controls?

PLOS authors have the option to publish the peer review history of their article (what does this mean?). If published, this will include your full peer review and any attached files.

Reviewer #1: No

Reviewer #2: Yes: Allan J. Guimaraes
---

## [Editor Report · Decision Letter 1]

12 Aug 2024

Dear Professor Borges,

Thank you for your robust response to the reviewer comments on the prior version of the manuscript. We are pleased to inform you that your manuscript 'Paracoccidioides lutzii Infects Galleria mellonella Employing Formamidase as a Virulence Factor' has been provisionally accepted for publication in PLOS Neglected Tropical Diseases.

Best regards,

Joshua Nosanchuk, MD

Section Editor

---

## [Editor Report · Acceptance letter]

25 Aug 2024

Dear Professor Borges,

We are delighted to inform you that your manuscript, "Paracoccidioides lutzii Infects Galleria mellonella Employing Formamidase as a Virulence Factor," has been formally accepted for publication in PLOS Neglected Tropical Diseases.

Best regards,

Shaden Kamhawi

co-Editor-in-Chief

Paul Brindley

co-Editor-in-Chief
